# A PointNet-Based Solution for 3D Hand Gesture Recognition

**DOI:** 10.3390/s20113226

**Published:** 2020-06-05

**Authors:** Radu Mirsu, Georgiana Simion, Catalin Daniel Caleanu, Ioana Monica Pop-Calimanu

**Affiliations:** Applied Electronics Department, Faculty of Electronics, Telecommunications and Information Technologies, Politehnica University Timișoara, 300223 Timișoara, Romania; georgiana.simion@upt.ro (G.S.); catalin.caleanu@upt.ro (C.D.C.); ioana-m.pop@upt.ro (I.M.P.-C.)

**Keywords:** PointNet, time of flight sensors, hand gesture recognition

## Abstract

Gesture recognition is an intensively researched area for several reasons. One of the most important reasons is because of this technology’s numerous application in various domains (e.g., robotics, games, medicine, automotive, etc.) Additionally, the introduction of three-dimensional (3D) image acquisition techniques (e.g., stereovision, projected-light, time-of-flight, etc.) overcomes the limitations of traditional two-dimensional (2D) approaches. Combined with the larger availability of 3D sensors (e.g., Microsoft Kinect, Intel RealSense, photonic mixer device (PMD), CamCube, etc.), recent interest in this domain has sparked. Moreover, in many computer vision tasks, the traditional statistic top approaches were outperformed by deep neural network-based solutions. In view of these considerations, we proposed a deep neural network solution by employing PointNet architecture for the problem of hand gesture recognition using depth data produced by a time of flight (ToF) sensor. We created a custom hand gesture dataset, then proposed a multistage hand segmentation by designing filtering, clustering, and finding the hand in the volume of interest and hand-forearm segmentation. For comparison purpose, two equivalent datasets were tested: a 3D point cloud dataset and a 2D image dataset, both obtained from the same stream. Besides the advantages of the 3D technology, the accuracy of the 3D method using PointNet is proven to outperform the 2D method in all circumstances, even the 2D method that employs a deep neural network.

## 1. Introduction

Gesture recognition has numerous applications: human–computer interaction (HCI), human–robot interaction (HRI), video surveillance, security, sports, and more. Recently, three-dimensional (3D) sensors based on the time of flight (ToF) principle emerged as a promising technology with clear advantages over two-dimensional (2D) approaches. These sensors are: (1) non-intrusive, since only depth data could be collected; (2) include an ambient illumination invariance used in low light or complete darkness; and (3) feature a simple segmentation process [1].

Regarding the principles used for 3D hand gesture recognition, one could divide them into two broad classes: (1) engineered features, extracted from 3D data and (2) implicit features, extracted automatically using a deep neural network.

A variety of handcrafted features are used in hand gesture recognition, some of which are briefly presented in the following paragraphs. One approach uses skin color to detect and segment the hand [2] and obtain binary silhouettes. They are further normalized using gesture geometry and Krawtchouk moment features, which are argued to be robust viewpoint changes, in order to classify the gesture. Konecny et al. [3], combined the appearances of histograms of oriented gradients and motion descriptors. Histograms of optical flow with dynamic time warping (DTW) simultaneously performed temporal segmentation with recognition. Wu et al. [4], from RGB and depth sequences, extracted motion descriptors based on the extended motion history image (extended-MHI), where recognition is performed using a maximum correlation coefficient. Wan et al. [5] proposed mixed features around sparse key points (MFSK), including spatiotemporal features extracted from RGB-D data, which has proven to have rotation and partial occlusions, as well as being robust and invariant to scale. Features like PCA, LDA, SVM, and other classifiers were used for further gesture recognition. Tamrakar et al. [6] evaluated several low-level appearances (e.g., SIFT and colorSIFT) as well as spatiotemporal features (dense trajectory-based HoG, MoSIFT, and STIP) using bag-of-words (BoW) descriptors and support vector machine approaches for gesture recognition. In some works, dense trajectory features are used [7] in combination with the Fisher vector encoding method instead of a bag-of-words histogram. Ohn-Bar et al. [8] described a system for hand gesture recognition performed in vehicles. RGB and depth information is used. The hand is detected in an ROI. In the next stage, the gesture is classified.

Unfortunately, the handcrafted feature-based methods cannot take all factors into consideration at the same time. In contrast, there is a growing trend represented by deep neural network (DNN), which have demonstrated remarkable performances on the large-scale and challenging gesture dataset [9]. A brief overview of deep learning (DL) and DNN solutions for hand gesture recognition is given below.

The idea of applying 3D CNNs on voxelized shapes represents one of the first deep learning techniques when dealing with 3D data, though this kind of representation is constrained by its resolution [10,11,12]. Molchanov et al. [13] proposed a 3D convolutional neural network classifier consisting of two sub-networks: a high-resolution network and a low-resolution network for drivers’ hand gesture recognition derived from both depth and intensity data, so that the inputs in the gesture classifier contained interleaved gradients of intensity frames and depth frames [13]. The final decision is obtained by multiplying the class-membership probabilities from the two sub-networks. They report a classification rate of 77.5% on the VIVA dataset. The principle was further extended by Molchanov et al. [14] by adding a recurrent layer for global temporal modeling. The dynamic gesture is split into multiple non-overlapping clips. For each clip, a 3D CNN structure extracts features based on a connectionist temporal classification cost function. The method yields top results, e.g., 98.6% for the SKIG RGB-D gesture dataset and 98.2% for the Chalearn 2014 dataset. A similarly approach was proposed by Zhu et al. [15], but this time a multimodal gesture recognition was employed. The inputs are both RGB and depth frames, each of which are processed by a 3D CNN followed by a convolutional LSTM, spatial pyramid pooling and fully connected layer. A multimodal fusion generates the final scores. The proposed multimodal gesture recognition method demonstrates the state-of-the-art accuracy of 98.89% for the SKIG dataset and 51.02% for the IsoGD/ChaLearn validation set. The work was further improved by Zhang et al., who included an attention mechanism in ConvLSTM [16]. Kingan et al. [17] made use of a bottom-up top-down structure to learn attention masks in order to select meaningful point cloud data points. By concatenating a sequence of frames for point clouds, their neural network learns spatiotemporal representation, enabling them to obtain 94.2% accuracy using a Japanese gesture dataset. Other interesting ideas are proposed by Ge et al. to cope with depth images for hand pose estimation. These ideas include: (1) projecting the depth image onto three views and applying multi-view CNNs to regress three views’ heat-maps [18]; (2) encoding the 3D point cloud of the hand after the hand is segmented from the depth image as volumes storing the projective directional truncated signed distance function values are fed into a 3D CNN with three 3D convolutional layers and three fully-connected layers [19]; (3) using PointNet architecture to directly process the 3D point cloud [20]. Aghbolaghi et al. provides an excellent review concluding that deep learning approaches for gesture and action recognition could be included in one of the four non-mutually exclusive groups: (1) 2D CNNs used to process one or more frames from the whole video; (2) 2D motion features applied further to 2D CNNs, (3) 3D CNN models, e.g., 3D convolutions and 3D pooling; and (4) temporal models in which 2D or 3D CNNs are combined with temporal sequence modeling techniques (e.g., RNN, LSTM) [21]. The above methods are summarized in Table 1.

This work is based on the PointNet architecture proposed by Qi et al. [22], which has emerged as a general framework for point cloud data processing for applications like 3D object classification and part and semantic segmentation. The authors reported state-of-the-art results using ModelNet40 shape classification benchmark. Further architectural extensions include PointNet++ [23], Temporal PointNet [24] for one-shot learning, and Dual PointNet for fusing information from two depth cameras [25].

Not much work has been done in deep learning on point sets, in general, and, in particular, in employing PointNet architecture for the specific problem of 3D static gesture recognition. The major contributions of this paper are summarized as follows: (1) 3D ToF gesture dataset creation made publicly available on Kaggle [26]; (2) precise hand-forearm segmentation algorithm using principal component analysis (PCA), which provides very good data normalization; (3) normalized point cloud size using histogram-based decisions with respect to point neighborhoods; (4) direct 3D data (point cloud) provided to a deep neural network in the form of the PointNet architecture, for ToF static gesture recognition; (5) extensive experiments that show above a 95% recognition rate, surpassing a typical 2D deep convolutional neural network approach.

The remainder of this paper is organized as follows. Section 2.1 presents the process of dataset creation. Section 2.2 presents the hand multi-stage segmentation method that isolates the hand from the rest of the scene. Section 2.3 presents how the palm’s point cloud data is determined. The employed deep neural network architectures are referred in Section 2.4. Section 3 demonstrates the experimental results. The conclusion and future research possibilities are provided in Section 4.

## 2. Materials and Methods

### 2.1. The Dataset

The 3D dataset was produced by our research team at Politehnica University Timisoara and used a camera system based on the ToF principle [27]. In such a system, a modulated optical signal illuminates the scene and a photonic mixer device (PMD) sensor detects the reflected light. The distance to the measured point depends on the phase shift between the original and reflected signal. As a result, a depth map of the measured scene was produced. The x and y coordinates of each point were also made available. However, this is not a full 3D model of the scene as only the surface closest to the camera was available. The hardware used was PMD (vision)^®^ CamCube 3.0, which is a 3D camera produced by PMD Technologies (Siegen, Germany). The resolution of the sensor was 200 × 200, yielding a total of 40,000 measured points. A few thousands points were given as input to the neural network. Therefore, the resolution of the camera was enough and after segmentation the number of points remaining in the hand region matched the typical input size dimension of the PointNet architecture. Figure 1 presents an example of the measurement produced by the camera. The image uses a color map to represent the measured distances, where red represents closer points.

In addition to the 3D sensor, the PMD CamCube 3.0 also contained a 2D grayscale image sensor calibrated with a 3D sensor. As a result, the database contained sets of 3D/2D scene recordings. In each set, the 3D depth image was aligned to the 2D image so coordinates matched. The database consisted of six hand gestures performed by 10 different people. Each person moved their arm randomly while holding a hand gesture resulting in 20 different frames per gesture. In each frame the same gesture had different orientations and scale. As a result, the dataset consisted of 1200 hand gestures. The dataset was made publicly available [26].

### 2.2. Hand Multi-Stage Segmentation

In the segmentation part, a mixture of depth information and 2D spatial information was used. Figure 2 presents a flowchart of the proposed segmentation algorithm. Steps 2.2.1–2.2.3 were detailed in our previous work [28]. In this work, the classification of the hand gestures was done using a DNN that processed directly to the 3D point cloud. Therefore, the initial procedure needed to be enhanced with supplementary steps as presented below.

#### 2.2.1. Filtering 

Segmentation starts by smoothing the depth image to remove the speckle noise. This noise was a result of random fluctuations in the reflected signal, due to particularities of surfaces in the scene. The speckle noise was removed using a 3 × 3 median filter. Because the resolution of the camera was reduced, the 3 × 3 kernel size outperforms larger sizes by having the least negative impact on the signal. The results are presented in Figure 3a,b.

The above steps were repeated until the condition was met. After the hand volume was determined, all the points inside the volume were considered of interest; the rest were discarded. Figure 4a,d shows examples when the hand volume contained the hand points but also points belonging to other body parts.

#### 2.2.2. Finding the Hand Volume of Interest

An initial assumption was that the hand was the closest object in the scene. Therefore, points with small depth distances had a high probability of belonging to the hand. Because the hand faced the camera most times, the volume of interest—called hand volume—was defined containing all points with depths within 7 cm starting from the closest point.

The algorithm started with the closest point from the camera in the data and set the margin of the hand volume there. In order to detect possible outliers resulted from faulty measurements, the number of points inside the hand volume was counted. If the count did not exceed a threshold, the closest point was considered an outlier and the algorithm moved the margin of the hand volume to the next closest point.

#### 2.2.3. Clustering Inside the Hand Volume

In order to perform clustering, a binarized image of the data was used (see Figure 4b,e). A set of properties like area, centroids, orientation, and pixel list were calculated and, in the end, one or more clusters were determined. In the simplest case, a single cluster determines the data that belongs to the hand. If more than one cluster is present for right hand performed gestures, the upper left-most cluster is considered to belong to that hand. Figure 4c,f presents the selected cluster.

#### 2.2.4. Hand-Forearm Segmentation

The above procedure assured that the hand would be segmented from the rest of the body and other objects presented in the scene. However, depending on the angle between the arm and the depth dimension, a variable amount of data belonging to the forearm remained in the acquired 3D data. At later stages, the data underwent a scaling process. A variable amount of forearm data affected the representation of the palm/hand data. In the following paragraph, we propose a method that successfully separates the hand data from the forearm data.

The approach was based on the observation that the displacement of data around the symmetry axis was generally larger in the hand region and was separated from the forearm by a narrow region that represented the wrist. A principal component transformation (PCA) was applied on the hand cluster data. If the forearm was present in the dataset, the arm had an elongated shape and so the principal component would be along the symmetry axis of the arm, as seen in Figure 5a. However, if the forearm was already removed by the previous segmentation and if the data was spread along both dimensions, the principal component might not find the symmetry, as in Figure 5c.

In order to detect such cases, a ratio between the spreads along the two new axis was calculated:(1)$$R=\frac{\sum x_{\mathrm{PCA}}\left(k\right)}{\sum y_{\mathrm{PCA}}\left(k\right)}$$
where xPCA and yPCA represent the point coordinates after PCA transformation. If this R ratio was below 1.5, segmentation was not continued. The above threshold was found empirically.

As shown in Figure 6, two sliding windows were used to calculate a point-index based function called segmentation function Fseg, which is computed as follows. The first window was of variable size and searched for the area representing the hand. The second window was of fixed size and aimed at the detection of the wrist. At each step, the size of the first window increased, accumulating more points in the direction of the principal axis. Moreover, at each step, the value of the segmentation function was calculated as the ratio between the mean displacement of points along the second axis in the wrist window with respect to the hand window, using Equation (2).
(2)$$F_{\mathrm{seg}}=\frac{\sum_{\mathrm{Window}2}y_{\mathrm{PCA}}\left(k\right)}{\sum_{\mathrm{Window}1}y_{\mathrm{PCA}}\left(k\right)}$$

Examples of segmentation functions are shown in Figure 7. The index/variable of the segmentation function (*x*-axis) corresponds to the index of the point inside the data where the hand/wrist separation was done. After the segmentation function was determined, the algorithm presented in Figure 8 determined the segmentation position inside the segmentation function. The examples shown in Figure 7 show the segmentation functions together with the result obtained by applying the segmentation algorithm. The algorithm searches for the first local minimum of the function. In many cases, such as Figure 7c–e, the first local minimum was also the global minimum and allowed facile segmentation of the forearm at this index.

In other cases, the global minimum differed from the first local minimum (Figure 7a,b,f). In these cases, the segmentation position was gradually updated if the newly found minimum sufficiently differed from the previously considered position. Threshold TH was used for this comparison. The value of TH was chosen by performing a search between 0 and 1 and comparing the segmentation result of the algorithm to a set of 50 manually segmented examples. The optimal value of TH was found at 0.9, as this value best fit the geometrical particularities of the data.

Figure 9 shows the graphical representation of the segmentation results. Examples for all six gestures are presented. The examples correspond to the segmentation functions presented in Figure 7a–f.

In some situations (according to Figure 8), segmentation was not performed. Figure 10 and Figure 11 show such cases together with their segmentation functions. In Figure 10a–c, the global minimum, which was significantly different from any other minimums, was at the end of the data, so no segmentation was performed. In fact, no segmentation was performed when the segmentation index was within the last 5% of the dataset. In Figure 10d, no segmentation was performed, as explained above and shown in Figure 5c.

### 2.3. Creating the Point Cloud

#### 2.3.1. Point Cloud Size

As the recognition system employed a further neural network with fixed dimension input, the size of the point set needed to be brought to the same size as the net input. This was done either by sampling the point set if its size was larger than the net input or by replicating points if its size was smaller. To ensure that this process did not change the point set size by a significant amount, the size of the net input was chosen such that it best fit the average value of the point set size. This value was set to 1024. When deciding which points to remove or replicate, the following analysis was made. The points in the set were not considered of equal importance, as points near the edge of the hand were more shape descriptive and contained more information. For every point in the set, we counted the number of neighbors that contained data. For this purpose, a variable size kernel was used. The size of the kernel increased until at least one background point was found. Then, we counted the number of hand data points in the kernel. The result of the count was represented as a histogram. Points that have most neighbors were in the center of the data bulk and therefore were the best candidates to remove. After one point was removed, neighbors were recounted. In this way, we avoided removing several points from the same region. Figure 12 illustrates the chosen points to be removed. When points needed to be replicated the same procedure was used. The only difference was that in this case the candidate points had the smallest number neighboring data points (Figure 13). In both cases, choosing the candidate points were histogram-based decisions.

#### 2.3.2. Point Cloud Scaling

After the number of points was brought to the desired size, the point cloud were generated. Because the proximity of the hand to the 3D camera affected the scale of the data, all datasets (point clouds) were scaled such that they fit into a unit radius sphere. This step revealed the importance of a good segmentation, as any non-relevant data (forearm, outliers) affected the scale of the relevant data. Figure 14a–d shows a point cloud example at different orientations.

### 2.4. PointNet Deep Neural Network as Gesture Classifier

In the following, the new point cloud processing concept of PointNet was applied to 3D data for a hand gesture recognition application. As presented in Section 1, the 3D data format had numerous advantages compared to 2D image data. Moreover, the accuracy obtained with PointNet over 3D data was compared to the accuracy obtained using a convolutional deep neural network (CNN) for the same set, represented as 2D data.

#### 2.4.1. PointNet Architecture 

PointNet was introduced in 2016 by Qi et al. [22] as a solution to directly processing 3D point cloud data, without any additional transformations. The input is represented by a set of 3D points {Pi|i = 1, …, n}, where n is the number of points in the set. Each point was represented by its (x, y, z) coordinates. The network had three important properties:Invariance to set order. The outcome of the network should not depend on the order of the points in the set allowing any possible permutation amongst them.Locality information. Even though points are processed independently, the local information brought by neighboring points or structures should not be lost.Invariance to transformations. Transformations like rotations or translations should not affect the outcome of the network.

The above architecture was further enhanced in Qi et al. [23]. 

#### 2.4.2. CNN Architecture

Further, a CNN was proposed with the structure of the network presented in Table 2. There were several convolutional layers using 3 × 3 filters. The choice of this typical kernel size was justified by the reduced memory usage and faster computational workload. The number of filters/layer gradually increased while the max pooling layers reduced the data dimensionality. The complexity and depth of the network was chosen in accordance with the available (size, number) 2D gesture data. The next section provides details regarding the procedures for obtaining the 2D set from the 3D one.

## 3. Results

### 3.1. Network Training and Testing

The dataset presented in Section 2.1 is used for training and testing. The total number of samples was 1200 but only 1091 samples were used in the experiments. The rejected samples contained bad/outlier pixels as a result of noise being present in the acquisition process. The data was segmented and then converted to point cloud; the size of each point cloud set was 1024 points. Each set was scaled to fit in a unit radius sphere. Moreover, during segmentation, an equivalent 2D image dataset was produced. The information in this image 2D dataset was equivalent to the 3D point cloud database as it was simultaneously acquired using a grayscale camera. The 2D images were aligned to the 3D data and so the same segmentation coordinates were used.

The database was divided into training set and testing set. The following split percentages for train and test have been used: 50/50, 60/40, 70/30, 80/20, and 90/10. Further, two experiments were done for a batch size of 32 and 16. For all cases, 100 epochs were used for training, using an initial learning rate of 0.001 for an Adam optimizer.

### 3.2. Experiments

The simulations were performed using the following system: Intel^®^ Core ™ i5—7500 CPU @ 3.4 GHz, 16GB Ram, 64bit system (Intel, Santa Clara, USA), GPU: NVIDIA GeForce GTX1050 1.3 GHz, 2 GB RAM, 640 CUDA Cores (Nvidia, Santa Clara, USA). The software framework used for simulating PointNet was Tensorflow 1.13 (Google, Mountain View, USA), while the CNN implementation was carried out using Keras 2.2 (MIT, Boston, USA).

Figure 15 and Figure 16 show the obtained results for batch 32 and 16. For every train/test split percentage, 10 different datasets were generated by randomly shuffling the original dataset. Training and testing simulations were done for each case. The displayed accuracy was the mean value of the obtained accuracies for each case. The results show that the PointNet (3D data) outperformed the CNN (2D data) approach in all situations, with a recognition rate at least 95%. When the batch size was set to 32, the accuracy difference between the two approaches was larger. In all cases, the accuracy was better if more training examples were available (larger split percentage in favor of training). In all cases, the testing accuracy was correlated with training accuracy, showing that the network generalized well and did not suffer from overfitting. A slightly better accuracy for testing compared to training was the result of the regularization techniques used during training.

It is important to notice that the CNN (2D) approach used an image database segmented using the 3D segmentation method. Therefore, the 2D image database was segmented very well. If a 2D segmentation method was used instead, it would have most likely had worse results and so the overall 2D classification accuracy would not be as good. This is another argument in favor of the 3D approach.

## 4. Conclusions

This paper successfully demonstrated the efficiency of a PointNet-based approach for the problem of 3D ToF static gesture recognition. Future interests are to investigate the performance and possibilities brought about by the novel point-voxel CNN introduced in Liu et al. [29]. Moreover, evaluating the accuracy of this new network applied to the gesture application presents interest. The computational workload and memory constrains associated to these models need to be determined, especially during inference, in an endeavor to find efficient models that can be deployed and then run in real time, on embedded systems, with limited resources.

## Figures and Tables

**Figure 1 sensors-20-03226-f001:**
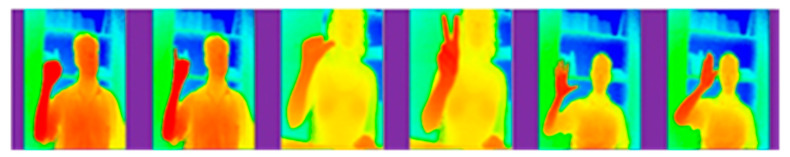
200 × 200 depth image acquired using a photonic mixer device (PMD) CamCube 3.0 three-dimensional (3D) time of flight (ToF) camera. Color map used: red—close; blue—away.

**Figure 2 sensors-20-03226-f002:**
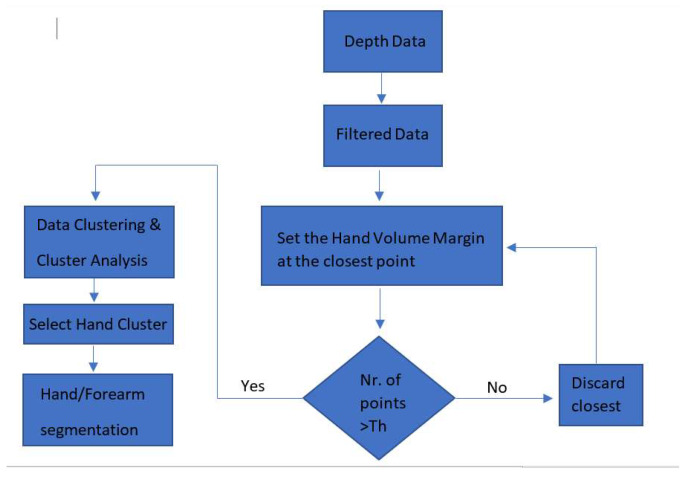
Multistage segmentation algorithm flow chart.

**Figure 3 sensors-20-03226-f003:**
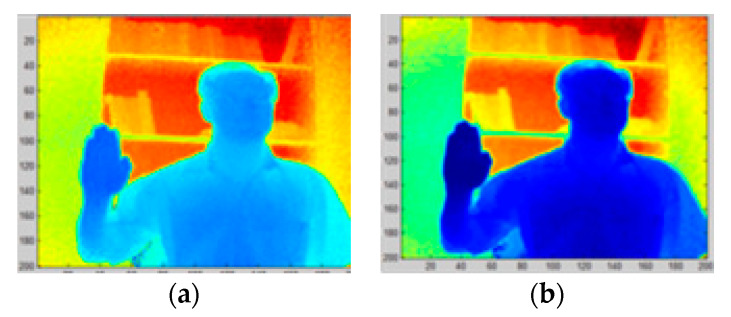
Depth image filtering using a 3 × 3 median filter: (**a**) original depth image; (**b**) filtered depth image.

**Figure 4 sensors-20-03226-f004:**
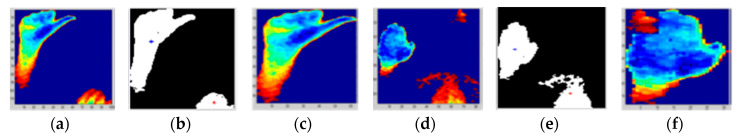
Finding the hand volume of interest. Clustering. Selecting the correct hand cluster. (**a**,**d**) original depth image. (**b**,**e**) binarized images. (**c**,**f**) selected cluster

**Figure 5 sensors-20-03226-f005:**
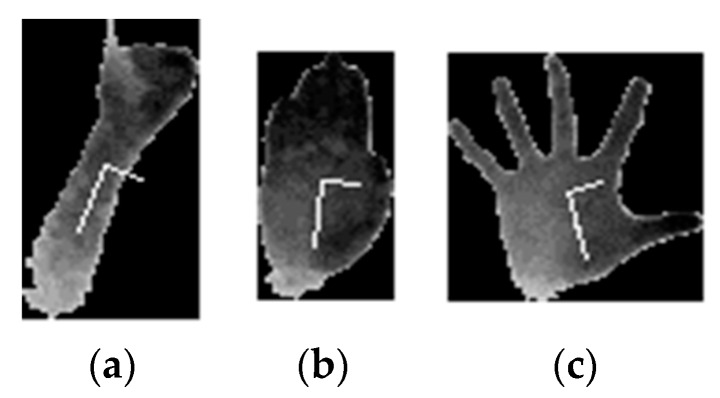
Principal component transformation (PCA) finds new axis aligned to the symmetry axis of the hand-forearm. (**a**) PCA axis aligned from forearm presence; (**b**) PCA axis aligned due to gesture shape; (**c**) PCA axis not aligned.

**Figure 6 sensors-20-03226-f006:**
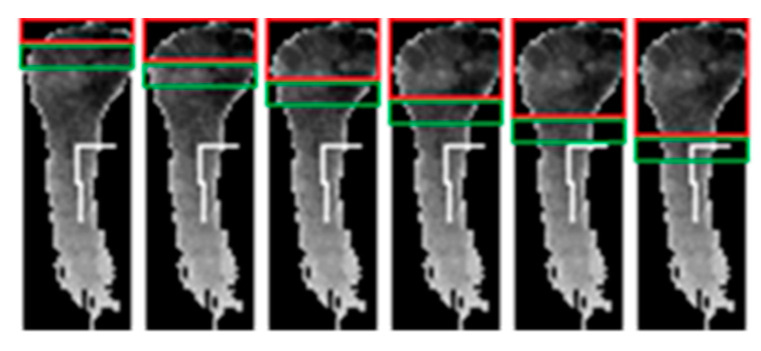
Two sliding windows were used to calculate the segmentation function. Window 1 (red-variable size), used for detecting the hand. Window 2 (green-fixed size), used for detecting the wrist.

**Figure 7 sensors-20-03226-f007:**
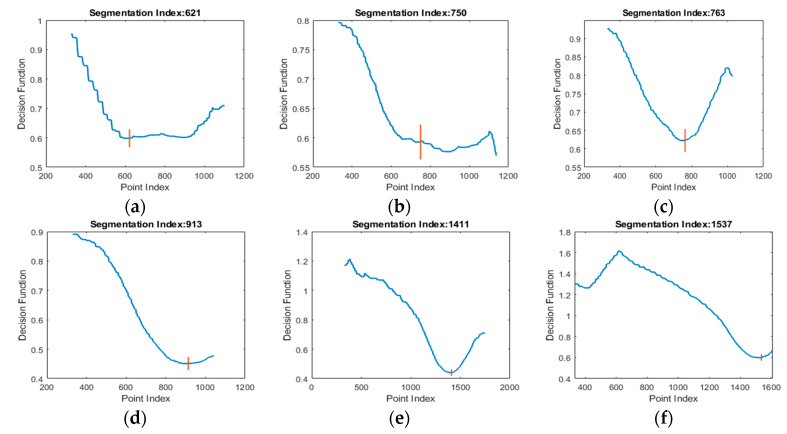
Examples of segmentation functions and the position of the segmentation index obtained by applying the segmentation algorithm. In (**a**,**b**,**f**) the minimum is gradually updated. In (**c**–**e**) local minimum is the same as global minimum.

**Figure 8 sensors-20-03226-f008:**
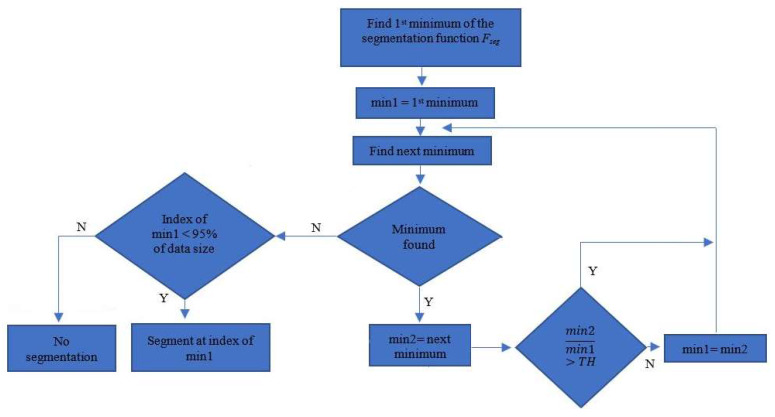
Segmentation algorithm.

**Figure 9 sensors-20-03226-f009:**
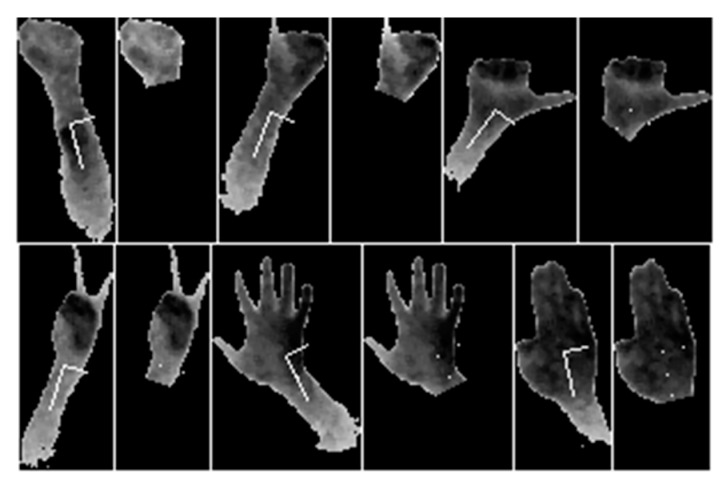
Examples of segmentation result (all six gestures).

**Figure 10 sensors-20-03226-f010:**
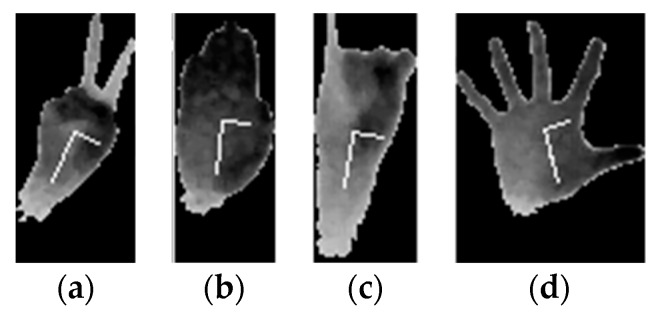
Examples when no segmentation is performed. (**a**–**c**) have a segmentation index within the last 5% of the dataset. (**d**) PCA axis not aligned as in Figure 5c.

**Figure 11 sensors-20-03226-f011:**
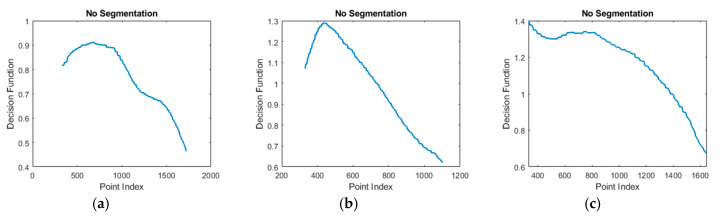
Segmentation functions. (**a**) corresponds to image in Figure 10a, (**b**) corresponds to image in Figure 10b, (**c**) corresponds to image in Figure 10c.

**Figure 12 sensors-20-03226-f012:**
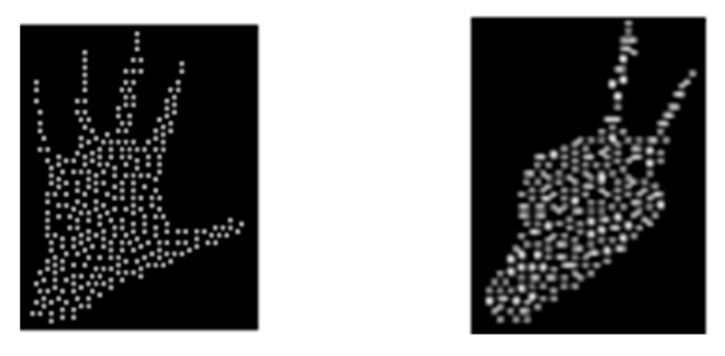
Removing points from the point cloud. Points with the largest number of neighbors are considered the best candidates.

**Figure 13 sensors-20-03226-f013:**
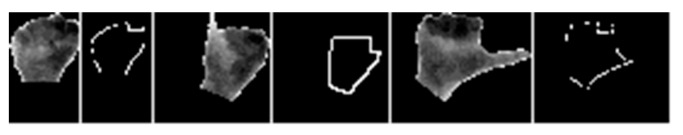
Replicating points from the point cloud. Points with the smallest number of neighbors are considered the best candidates.

**Figure 14 sensors-20-03226-f014:**
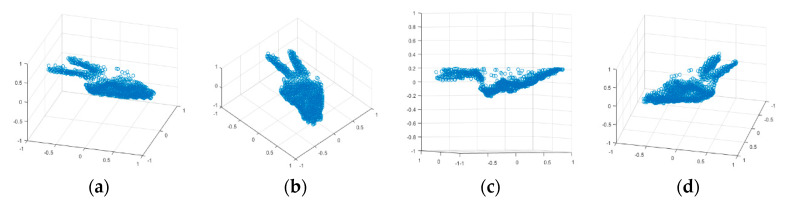
Point cloud example of size 1024. Data scaled to fit in a unit radius sphere. (**a**)—orientation 1, (**b**)—orientation 2, (**c**)—orientation 3, (**d**)—orientation 4

**Figure 15 sensors-20-03226-f015:**
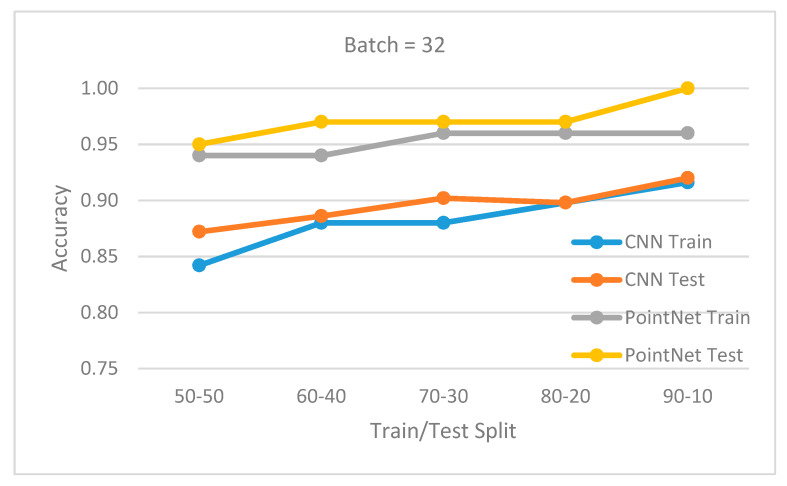
Batch 32. Mean accuracy for different train/test splits.

**Figure 16 sensors-20-03226-f016:**
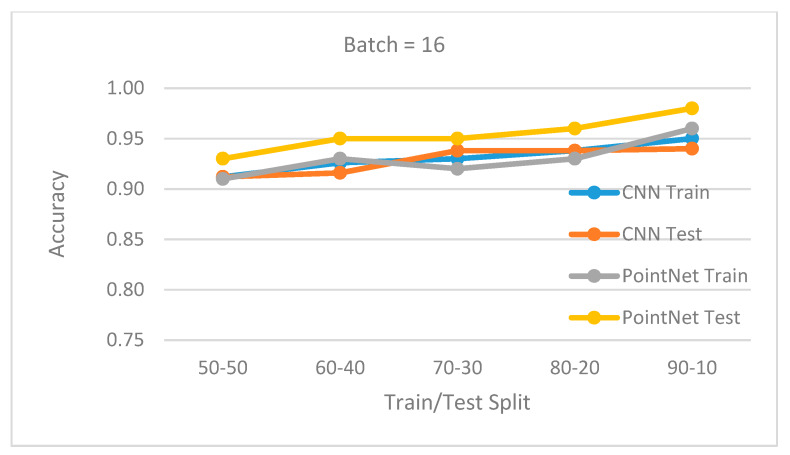
Batch 16. Mean accuracy for different train/test splits.

**Table 1 sensors-20-03226-t001:** An overview of deep neural networks (DNN) for gesture recognition.

Method	Method Summary	Dataset	Accuracy %
Molchanov et al. [13]	CNN, 2 sub-networks	Viva Challenge Dataset	77.5
Molchanov et al. [14]	Recurrent 3D CNN	SKIG/ChaLearn 2014	98.6/98.2
Zhu et al. [15]	3D CNN + LSTM	SKIG/ChaLearn (Lap IsoGD)	98.89/52.01
Zhang et al. [16]	ConvLSTM	Jester/IsoGD	95.13/55.98
Kingan et al. [17]	PointNet + Attention Module	Japanese Gesture Dataset	94.2
Ge et al. [18,19,20]	3D CNN	MSRA/NYU/ICVL	-

**Table 2 sensors-20-03226-t002:** Layer architecture of CNN.

Layer (Type)	Output Shape	Param#	Layer (Type)	Output Shape	Param#
Conv2d_1 (Conv2D)	(62, 44, 32)	320	Conv2d_5 (Conv2D)	(14, 9, 128)	73,856
Conv2d_2 (Conv2D)	(60, 42, 32)	9248	Conv2d_6 (Conv2D)	(12, 7, 128)	147,584
max_pooling2d_1 (MaxPooling2)	(30, 21, 32)	0	max_pooling2d_3 (MaxPooling2)	(6, 3, 128)	0
dropout_1 (Dropout)	(30, 21, 32)	0	dropout_3 (Dropout)	(6, 3, 128)	0
Conv2d_3 (Conv2D)	(30, 21, 64)	18,496	flatten_1 (Flatten)	(2304)	0
Conv2d_4 (Conv2D)	(28, 19, 64)	36,928	dense_1 (Dense)	512	1,180,160
max_pooling2d_2 (MaxPooling2)	(14, 9, 64)	0	dropout_4 (Dropout)	512	0
dropout_2 (Dropout)	(14, 9, 64)	0	dense_2 (Dense)	6	3078
Total params:	1,469,670

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
