# Peer review of "A PointNet-Based Solution for 3D Hand Gesture Recognition"

_sensors, 2020, doi:10.3390/s20113226_

Round 1
Reviewer 1 Report
The manuscript presents a novel method for analysing 3D captured hand gestures images using PointNet and machine learning to determine the gestures themselves. The manuscript on the whole is well written and the methodology is well presented. I suggest improvements to the CNN training strategy described in later sections to ensure more robust conclusions are drawn about accuracy and regularisation.
General Comments
- English is of a high standard, some minor errors found so suggest a thorough re-read
- Change sentences to remove personal pronouns like "our" such as in line 114, line 140 and line 250 and "we" in line 282 (check for other cases too)
Introduction Section
Overall the section is well written, well referenced and has a clear structure. A suggestion for ease of comparison of the deep learning solutions from line 59 onwards could be a table to summarise the performance of each (name[ref], summary of method, accuracy).
Materials and Methods
--- The Dataset
Line 115 - is there an open access site with the data yet or will that be part of the publishing process with MDPI?
Line 123 "therefore the resolution...is satisfactory" - how was this assessed? Is there any mention in the literature reviewed in the Introduction of the resolution used in those applications? Whilst in Figure 1, it seems satisfactory as you are able to label images yourself, some qualification of the statement is needed.
--- Filtering
Line 148 "easily removed" - use more formal language here and qualify such as "As speckle noise is broadly stochastic spatially throughout the image, median filtering can be used to reduce it's impact." You don't have to use those exact words, just improve and also qualify the use of of a 3x3 filter, for example would 5x5 be better or starts to impact signal?
--- Finding the Hand Volume
Line 153 - How was the shallow volume determined? Is 7cm the radius of a circle in a 2D plane from the point closest to the camera? What if the palm is closest, would this radius miss the fingertips? Figure 4 shows the last stage of discarding, but some illustration of the initial stages of selecting this 7cm region is needed.
--- Hand-Forearm Segmentation
Line 197 - Would kernel PCA help develop a non-empirically defined threshold for forearm/hand differentiation? The method the authors uses does seem quite effective, just wondering about transfer into other applications (like another hand gesture dataset), a new threshold would be calcluated each time?
--- Point Cloud Size
The explanation of how to reduce the number of points is good and the logic seems reasonable. Minor point, Figure 12 would be more illustrative if the comparison is between the original segemented image and the then the reduced points, rather than 2 reduced point images.
--- PointNet Architecture
Line 299 - Please don't suggest further reading to the reader! The authors should ensure the key information from cited sources is succinctly contained within the manuscript. The preceding text does seem to summarise this information, so either remove this sentence or expand section 2.4.1 to contain additional information about PointNet relevant for the application presented by the authors.
--- CNN Architecture
Line 302 - briefly explain why 3x3 filter is used.
--- Network Training and Testing and Experiments
The training methodology needs improvement. The authors state different train/test splits and 2 different batch sizes. I would suggest that a single train test split is used (70/30 or 80/20) and the different batch size is expanded (if possible) to 16, 32 and 64. The authors can then use k-folds cross validation - of the 70% or 80% of training instances available, these are further split into 10 groups which can be be trained and tested against each other. This will indicate how mcuh the mode is affected by extremes. It is also useful on smaller datasets when considering the number of samples. Authors used 1091 samples, were these roughly split equally across all the gesture types or was there an imbalance? These information is needed to have porper contest of the model performance.
Why was 100 epochs selected? Could you not look for local minima on the test plot of test/train curves for early stopping?
The authors state that using a batch size of 32 seems to improve performance as it increases regularisation. This is likely true, but only really justifiable with tests on larger and smaller batch sizes.
Reviewer 2 Report
The authors present a PointNet based for the 3D hand gesture recognition. They detailed the processing steps. There are some comments aimed to improve the quality of the paper:
- Figure 2 is not clear. Please re-draw them in high-resolution way.
- The format of Figure 10 is not correct. Please modify it based on journal format.
- For consistency, please insert the (a), (b), (c) and (d) in Figure 14 to represent from Figure 14(a) to Figure 14(d). Also, please add the description for Figure 14 to make this figure presentation more clear.
- The format of “Table 1” is not correct. Please modify it based on journal format.
- The Figure 15 and 16, Pleas insert the title of axes.
- The format of “References” is not correct. Please modify it based on journal format.
